# A Glycoproteinaceous Secretion in the Seminal Vesicles of the Termite *Coptotermes gestroi* (Isoptera: Rhinotermitidae)

**DOI:** 10.3390/insects10120428

**Published:** 2019-11-26

**Authors:** Lara T. Laranjo, Ives Haifig, Ana Maria Costa-Leonardo

**Affiliations:** 1Department of Biology, Laboratory of Termites, Institute of Biosciences, São Paulo State University—UNESP, Campus Rio Claro, 24A Avenue, 1515, Bela Vista, Rio Claro, SP 13506-900, Brazil; ltlaranjo@gmail.com; 2Center for Natural and Human Sciences, Federal University of ABC—UFABC, Building Delta, room 241, 03 Arcturus Street, Jardim Antares, São Bernardo do Campo, SP 09606-070, Brazil; ives.haifig@ufabc.edu.br

**Keywords:** Blattaria, kings, neotenics, reproductive system, spermatozoa, testes

## Abstract

*Coptotermes gestroi* is a subterranean termite with colonies generally headed by a pair of primary reproductives, although neotenics may occur. In this study, the male reproductive system was compared during different life stages of nymphs, alates, neotenic reproductives, and kings of *C. gestroi*, focusing on the modifications of this system along the maturation of these individuals. The structure of the male reproductive system follows the pattern described for insects, although *C. gestroi* males do not exhibit conspicuous penises and differentiated accessory glands. In kings, each testis consisted of about seven lobes, significantly increased in size as compared to younger males. The spermatogenesis begins in third-instar nymphs, which already presented spermatozoa in the testes. The seminal vesicles are individualized in *C. gestroi* and have a secretory distal portion and a proximal portion with a role in spermatozoa storage. The secretion of the seminal vesicles is strongly periodic acid Schiff (PAS)-positive, whereas the xylidine Ponceau test revealed proteins that increase in quantity while the males become older. This is the first record of glycoproteins in the lumen of seminal vesicles in termites. Further studies will clarify how they are produced and interact in the physiology and nutrition of the non-flagellate spermatozoa of *C. gestroi*.

## 1. Introduction

*Coptotermes gestroi* is a rhinotermitid native to southeastern Asia [1], and is responsible for severe damage and economic impacts on structural woods in Brazil [2]. The colony population of this termite reaches more than one million of individuals, and a colony is distributed in polydomous nests, which may be either subterranean or disposed above ground, usually in building basements or skyscraper plenums [3].

The royal couple of *C. gestroi* is housed in the principal nest, which spreads with colony growth in secondary structures named satellite nests. Replacement reproductives appear in colonies when the royal couple dies or during colony fission. These individuals are nymphoid neotenics because they have wing buds and are originated from nymphs. Non-functional neotenic reproductives seem to be precursors of mature functional neotenics and were already observed in colonies with kings and queens [4], but hitherto only one functional replacement queen was found in a nest located on the seventeenth floor of a skyscraper in São Paulo city [5].

Termite castes are determined during their post-embryonic development, and the hemimetabolous development from eggs to alates is characterized by larval and nymphal instars, which are immature stages in the imaginal line [6,7]. Barsotti and Costa-Leonardo [8] found six nymphal instars in *C. gestroi*, but only third-, fourth-, and fifth-instar nymphs are engaged in foraging activities [9]. According to [10], information regarding the reproductive system of nymphs is scarce compared to that available for the termite royal couple and alates.

Morphological variation of male reproductive apparatus and spermatozoa are found in social insects and within termite families [11,12,13]. Termites present reduced external male genitalia, usually not sclerotized or entirely absent [14,15], except for the species *Mastotermes darwiniensis* which presents a genital papilla [16] and *Stolotermes inopinus* which presents a phallic lobe [17]. The termite spermatozoa may be flagellate or aflagellate and present different sizes and shapes. The most basal family Mastotermitidae shows multiflagellate spermatozoa, but in other families flagella are absent and exhibit different shapes, such as a rod-like shape as in Hodotermitidae, a conical form or “zinnia seed” shape as in Kalotermitidae, or a spheroidal form as in Rhinotermitidae and Termitidae [12,13,18,19].

Costa-Leonardo and Barsotti [20] verified that the reproductive system of a male alate in *C. gestroi* is composed of two testes, two vasa deferentia, two seminal vesicles, and an ejaculatory duct, which is located under the 9th abdominal sternite. The testes are composed of several lobes grouped together and located latero-dorsally in the posterior part of the abdomen. In alates, the testes are densely grouped and not well-developed, but in a king of an incipient 2-year-old colony, seven distinct testicular lobes were described [3].

The present study investigated the morphological alterations of the reproductive system in males of *C. gestroi* during the latter post-embryonic development, from third-instar nymphs to kings. In addition, this study aimed to qualitatively analyze the seminal vesicles and their secretory function during the maturation of these reproductives, using histochemical and ultrastructural techniques. The development of the male reproductive system was compared through light microscopy before the imaginal molt, using latter nymphal instars and neotenics, after it, using alates, and in subsequent years after colony establishment, using kings of different ages.

## 2. Materials and Methods

### 2.1. Insects

Male alates of *Coptotermes gestroi* (Wasmann, 1896) were collected during dispersal flights. Functional primary kings, that is, individuals that establish the colonies with the queens, both as alates, were collected from laboratory colonies 6 months, 1, 2, 4 and 6 years after colony establishment from wild-caught dispersing alates. Male nymphoid neotenics were collected from a field nest which had a royal couple. Foraging nymphs were collected from field traps that were placed in urban areas. We used three individuals of each age for histology and histochemical analysis, except for alates and 1-year-old kings for which six individuals were used. For morphology, 21 alates, three 4- and one 6-year-old kings were dissected. For transmission electron microscopy, three alates and three 1-year-old kings were analyzed.

### 2.2. Morphology of the Reproductive System

Alates and 4- and 6-year-old kings of *C. gestroi* were dissected under a Zeiss Stemi SV6 stereomicroscope and had their reproductive organs isolated. Each reproductive organ was individually placed on a microscope slide and stained with a 1% methylene blue solution, and documented with the aid of a Motic-CAM camera (Causeway Bay, Hong Kong) using the Motic Image Plus 2.0 ML software. The total mountings were used for drawing a scheme of the male reproductive system of *C. gestroi*.

### 2.3. Histology and Histochemistry

Posterior abdomens of *C. gestroi* male reproductives were fixed in FAA (absolute alcohol, glacial acetic acid, 40% formaldehyde, in the proportion of 3:1:1) for 24 h. Later, they were dehydrated in graded series of ethanol concentrations (70%–95%), transferred to an infiltration resin solution (Leica^®^, Wetzlar, Germany) and embedded with historesin (Leica^®^) plus catalyzer for polymerization. After polymerized, 3-µm sections were obtained using Tungsten blades in a microtome (Leica^®^ RM2245). The histological sections were stained with Harris hematoxylin and aqueous eosin [21]. For the histochemical studies, sections of the abdomens of alates, neotenics and kings were stained with the following treatments:

Periodic acid Schiff (PAS) for detection of neutral polysaccharides (according to [21]): sections were immersed for 10 min in 0.4% periodic acid, washed with distilled water and stained with Schiff reagent for 1 h in the dark. The material was then washed three times with sulfur water (10 mL 10% sodium metasulfite: 10 mL 1N HCl: 180 mL distilled water) for 3 min each and rinsed with tap water for 30 min. After drying, slides were cleared with xylol and mounted in Canada synthetic balsam.

Alcian blue/PAS for acid and neutral polysaccharides detection (according to [22]): sections were stained with Alcian blue pH 2.5 for 30 min and washed in distilled water. Then, the material was transferred to 1% periodic acid for 5 min and washed. The sections were placed in Schiff reagent for 40 min, and washed in water for 10 min. After drying, slides were cleared with xylol and mounted in Canada synthetic balsam.

Xylidine Ponceau for total proteins detection [22]: sections were stained with xylidine Ponceau for 30 min and washed in distilled water. After drying, slides were cleared with xylol and mounted in Canada synthetic balsam.

The preparations were visualized using a photomicroscope (Leica DM500/Leica ICC50), with the aid of LAS v4.0 software (Leica Application Suite v4).

### 2.4. Morphometry of the Testes

A quantitative analysis of the testicular area using the three most sagittal sections of the testes, certified by the lumen of the vas deferens, from all of the individuals used in histology, except for 2-year-old kings, was performed using Image J 1.52 k software. Testicular area of the 2-year-old kings was not be determined from histological sections due to loss of some lobes during slide preparations. Data were log-transformed to achieve both homoskedasticity and normality and analyzed using one-way analysis of variance (ANOVA) followed by Tukey HSD test for multiple comparisons. The analyses were performed in the R program, v.3.6.0 [23].

### 2.5. Transmission Electron Microscopy (TEM)

Posterior abdomens of male alates and 1-year-old kings were fixed in 2.5% glutaraldehyde in 0.1 M cacodylate buffer (pH 7.0) and post-fixed in 1% OsO_4_ in cacodylate buffer. After fixation, the material was stained with uranyl acetate for 2 h. The samples were dehydrated through a graded series of acetone solutions (50%–100%), embedded in a solution of Epon Araldite (Araldite 501/PolyBed 812 kit, Polysciences, Germany) resin and acetone (1:1) for 1 h and then pure resin for 2 h. The material was embedded in resin with the catalyzer DMP 30 and polymerized at 60 °C for 24 h. Semi-thin sections (1.5–2 µm) were stained with methylene blue and azur II. Ultrathin sections (60–90 nm) were obtained using a Leica Reichert Supernova ultramicrotome and stained with 4% uranyl acetate and lead citrate. The material was observed under a CM100 Philips transmission electron microscope (operating at an accelerating voltage of 80 kV) and photographed with a Veleta camera using iTEM software (v. 5.2).

## 3. Results

### 3.1. Morphology of the Reproductive System

The reproductive system of males of *Coptotermes gestroi* is composed of two testes, two vasa deferentia, two seminal vesicles, and a common ejaculatory duct (Figure 1). The alates show conspicuous seminal vesicles, and testicular lobes appear as small and grouped structures, different from 6-year-old kings, in which the seminal vesicles are increased and the testicular lobes show a flower shape, being possible to distinguish one lobe from another. Five to nine testicular lobes are observed in 4- and 6-year-old kings (Figure 1B and Figure 2A).

### 3.2. Histology and Histochemistry

The increase in size of these testicular lobes is evident during the development of the reproductives of *C. gestroi* (Figure 2). Histological sections of the abdomens also expose a retractile penis in all male reproductives (Figure 2A). In nymphs and alates, it is not possible to distinguish each testicular lobe because they are slender and packed together (Figure 2B,D). The testes considerably increase in size and the germinative cell differentiation in cysts is observed in 6-month-, 1-, 2-, and 4-year-old kings (Figure 2E,H). In neotenics, testicular lobes are not so conspicuous as those observed in the kings, and present an intermediate developmental stage that can be included in a classification between alates and kings (Figure 2I). The testicular lobes in kings are composed of cysts of spermatids/spermatozoa, which are involved by individualized wall and are separated from one another by the peritoneal sheath. In each cyst, all germinative cells are at the same stage of development. In the seminiferous tubules, the cysts vary throughout a gradient formed from the apical to the basal region of the testes. The apical region of the testicular lobes presents groups of cells with rounded nuclei, whereas the cells of the basal region show chromatin with typical meiosis arrangements and spermatozoa (Figure 2). In *C. gestroi*, the spermatozoa are rounded, non-flagellated, and measure around 2 µm in diameter.

The vasa deferentia are paired ducts that leave ventrolaterally from the testes towards the base of the ninth sternite. These ducts are formed by an inner layer of simple epithelial tissue surrounded by a layer of circular muscle. Inside these ducts, spermatozoa are observed towards the seminal vesicles, where they are stored (Figure 2A).

The seminal vesicles are paired structures in all the reproductives (Figure 1). In nymphs, these structures are small and do not show spermatozoa in the lumen. However, these individuals already present spermatozoa in the testes since the third instar (Figure 2B), but they move to the seminal vesicles only in neotenics or after the imaginal molt, in alates. The seminal vesicles discharge at the final portion of the vasa deferentia, and have the appearance of two twisted tubes (Figure 1). In males of *C. gestroi*, the distal portions of the vesicles present a simple columnar epithelium while the proximal portions have a simple cubic epithelium and both are surrounded by musculature. The proximal region is easily distinguished due to the large amount of spermatozoa in the lumen (Figure 3). In addition, the smooth musculature in the proximal portion is thicker than that present in the distal portion (Figure 3A).

A characteristic secretion is present in the lumen of the seminal vesicles in all male reproductives (Figure 3B,F). The histochemistry shows that the secretion of the seminal vesicles is PAS positive in alates, neotenics, and 1- and 4-year-old kings (Figure 4A,B). However, the histochemical test of Alcian blue/PAS did not show acid polysaccharides as components of this secretion (Figure 4C). The PAS test also highlights the presence of a basal layer in the epithelium of the seminal vesicles both in proximal and distal regions in all examined individuals (Figure 4A,C).

Alates display a low volume of protein granules in the secretion present in the lumen of the seminal vesicles (Figure 4D), which progressively increases in number in 6-month-old kings and in 1-year-old kings (Figure 4E,F). Therefore, the results show a proteinaceous secretion in the lumen of seminal vesicles that is stored as a rounded secretion in younger kings but more homogeneously distributed in the seminal lumen in older kings.

### 3.3. Morphometry of the Testes

The testes significantly increase in size from the third-instar nymphs to older kings (ANOVA: F = 3036; df = 7, 16; *p* < 0.001). The testicular area did not vary between nymphs of third and fourth instars (Tukey: *p* = 0.0752) and between 6-month and 1-year-old kings (Tukey: *p* = 0.3518). The testicular area comparisons are given in Table 1.

### 3.4. Transmission Electron Microscopy (TEM)

Semi-thin sections of seminal vesicles evidence the smooth musculature involving the secretory epithelium of these organs (Figure 5A). The ultrastructure of the seminal vesicles in alates and kings shows many microvilli in the apical region of the epithelial cells (Figure 5B,C). A flocculated secretion is observed in the lumen of seminal vesicles (Figure 5C), whereas profiles of rough endoplasmic reticulum are observed in the cytoplasm of the epithelial cells (Figure 5D). In both alates and kings, a thick basal lamina is surrounded by visceral muscle cells (Figure 5E), and septate junctions link adjacent epithelial cells (Figure 5F,G). The cytoplasm of these cells has rough endoplasmic reticulum, polyribosomes, Golgi apparatus, and few mitochondria.

Many strongly electron-dense spherical spermatozoa lie close to degenerative cells in the proximal lumen of the seminal vesicles (Figure 5H). The detail of a spermatozoon (ca. 2 µm diam) revealed a spherical electron-dense nucleus with an electron-lucent area (Figure 5I). The spermatozoa show, in one of their poles, a small region that houses two centrioles and two mitochondria. Microtubules were not observed in spermatozoa or in spermatids.

## 4. Discussion

The morphology of the reproductive system of *Coptotermes gestroi* is similar to that described by [15] for the termite *Reticulitermes hesperus*. Testicular lobes of functional kings of *C. gestroi* were completely separate from one another, as occur in Diptera and some Orthoptera, in which the testes consist of individualized structures that are connected to the vas deferens [24]. In this study, the 6-year-old king of *C. gestroi* presented up to nine lobes per testis. The number of testicular lobes varies among species, but in lower termites, there are generally 7 to 10 lobes per testis [15]. The number of testicular lobes was difficult to determine in alates, as the testes were not well developed and the lobes were compactly grouped in these reproductives. On the other hand, kings of *C. gestroi* presented testicular lobes perfectly separated in the form of fingers similar to those described by [25] for other males of the family *Rhinotermitidae*.

Nymphs and alates did not have isolated testicular lobes, and this morphological feature seems to differentiate functional and non-functional reproductives in a colony of *C. gestroi*. Neotenics presented an intermediate stage of testicular development between alates and kings. Then, the testicular lobes of *C. gestroi* grew from nymphs to 4-year-old kings, reaching the greatest size in the latter. Ye et al. [26] found testicular lobes of 5-year-old kings nearly two times larger than those present in unflow alates of *R. flavipes*. As kings of *C. gestroi*, kings of *R. hesperus* showed each testicular lobe with two distinct regions: an apical region composed by groups of squamous cells and a basal region with spermatids and mature spermatozoa inside cysts [15]. The spermatozoon of *C. gestroi* is spherical and lacks microtubules [3], very similar to that observed in the genus *Reticulitermes* [27] and different from other lower termites such as *Mastotermes darwiniensis, Kalotermes flavicollis*, and *Zootermopsis nevadensis*, which have those microtubular structures [28].

The spermatozoon production in the testicular lobes starts early in the post-embryonic development of *C. gestroi*, as observed in third-instar nymphs. Spermatozoa move from the testes to the seminal vesicles where they are stored before being transferred to the female genital apparatus during copulation. In the present study, spermatozoa were found in the proximal portion of the seminal vesicles in alates, neotenics and kings, and together with the testicular development, this feature might indicate the reproductive status of these individuals. In *Hodotermopsis sjostedti*, all castes, except young larvae, presented spermatozoa in the testes, but only reproductives, including nymphs, neotenics and alates showed sperm in the vas deferens [29]. Other studies also associated the reproductive status of the individuals with testicular development and the presence of sperm in well-developed seminal vesicles [30,31]. Our results suggest that the neotenics of *C. gestroi* might be functional, although they were collected from a colony in which the primary reproductives were present and the female neotenics did not present terminal oocytes and sperm in the spermathecae [4], because as observed for *H. sjostedti*, the male system development seems to be more accelerated when compared to the female system. According to [30], even before external anatomical modifications some male neotenics seem be functional in *R. labralis*, the so-called inconspicuous male neotenics.

Neotenic reproductives were reported in several species of *Coptotermes* [32], in which they are commonly differentiated from the fifth nymphal instar. Often, non-functional individuals occur in the presence of the primary reproductives, and become functional after colony orphaning [4,32]. In *Silvestritermes euamignathus*, spermatogenesis already occurs in third-instar nymphs, but spermatozoa were only observed in the testicular lobes of fifth-instar nymphs [33], latter than in *C. gestroi*. However, third- to fifth-instar nymphs of *S. euamignathus* may differentiate into neotenics after two successive molts [34], and these functional nymphoid neotenics present spermatozoa in the seminal vesicles [33].

The seminal vesicles of *C. gestroi* are individualized organs with morphological and probably functional differentiations in their different portions (proximal and distal). According to [24], the seminal vesicles are dilations of the vasa deferentia in many insects, such as observed in the higher termites, e.g., *S. euamignathus* [33] and some lower termites, e.g., *Cryptotermes brevis* [35,36]. However, in some orthopteroid insects, the seminal vesicles are not simple expansions of the ducts, but individualized structures as observed in *C. gestroi*. The distal portion of the seminal vesicles in *C. gestroi* has its lumen filled with secretion, whereas spermatozoa are found in the proximal portion. Therefore, the seminal vesicles are secretory structures in *C. gestroi*. Weesner [37] also found well-developed seminal vesicles with their lumen filled with a clear secretion containing some inclusions and few spermatozoa in *R. hesperus.*

According to [24], the cells of the seminal vesicles of some insects are glandular and probably provide nutrition for the spermatozoa. The seminal vesicles described for other lower termites, such as *Hodotermes mossambicus*, have a secretory epithelium formed by cells with canals (class 3 glandular cells according to [38,39]) and enveloped by a muscle layer, thicker in the proximal portion [40]. However, as these canals are cuticular in origin, if they were present in the seminal vesicles of *C. gestroi*, the PAS technique would have stained them in magenta, which did not occur, showing that these class 3 glandular cells are absent in *C. gestroi*. We also did not observe these cells in the ultrastructure, but the occurrence of microvilli and secretory vesicles in the epithelium indicate class 1 glandular cells (classification by Noirot and Quennedey [38,39]) in the seminal vesicles of *C. gestroi*.

All male reproductives, alates and kings of different ages, showed a secretion that was intensely stained with PAS in the lumen of the seminal vesicles. The histochemical test of xylidine Ponceau also confirms the occurrence of proteins in the seminal vesicle secretion. The positive PAS and xylidine Ponceau outcome corroborates the hypothesis that the secretion of seminal vesicles is composed of glycoproteins in *C. gestroi*, as these organs produce proteins or peptides in many insects [41]. An increase in the amount of proteins, showed by the xylidine Ponceau test, occurs during the maturation of the males. In addition, the results of PAS showed no polysaccharides in the cytoplasm of the epithelial cells of the seminal vesicles, but the glycoproteins present in the basal membrane of the epithelium were intensely stained.

The ultrastructure of the seminal vesicles also showed a high amount of rough endoplasmic reticulum in the cytoplasm of epithelial cells in male reproductives of *C. gestroi*, which is indicative of a high protein production by these cells. According to [42], proteins found in the seminal vesicles and in the accessory glands of insects have a range of functions that, collectively, serve to improve the reproductive success of males. This was empirically demonstrated in *Apis mellifera*, in which the seminal fluid proteins positively influenced sperm viability [43]. They may also be produced in other regions of the male reproductive system, such as the ejaculatory ducts, and are known as proteins of the seminal fluid. These proteins act in all phases of the reproductive biology of inseminated females and have several functions, such as reducing the receptivity or attractiveness of females, antimicrobial properties, improving the protection, storage, activation, and competition of spermatozoa, and inducing increased oogenesis and oviposition by females [42,43,44,45]. This is the first record of a proteinaceous secretion, which is probably composed of glycoproteins (protein-carbohydrate complex), in seminal vesicles of Isoptera. In the case of *C. gestroi*, these proteins may play all those roles, with especial emphasis on protection and storage of spermatozoa, as they are produced in the testes and stored in the seminal vesicles before being transported in a viable state to the female. Further studies will clarify how these glycoproteins produced in the male reproductive system improve the sperm storage and whether they affect the physiology and the behavior of termite females.

## 5. Conclusions

The reproductive system of *C. gestroi* males increases during the post-embryonic development, which is visible in the testicular lobes where the spermatogenesis occurs. In third instar nymphs, the spermatozoa are already present in these structures, showing that spermatogenesis starts before the individual becomes an adult. It is possible that this precocious sperm production may be related to the development of neotenic forms.

The testes of *C. gestroi* males continue to develop after the imaginal molt, during the aging of the kings, evidenced by the densely packaged testicular lobes in alates, which are greater and individually separated in kings.

The seminal vesicles of all male reproductives *C. gestroi* present two distinct parts, the proximal and the distal, related respectively to two distinct functions, the production of glycoproteinaceous secretion and the storage of spermatozoa.

## Figures and Tables

**Figure 1 insects-10-00428-f001:**
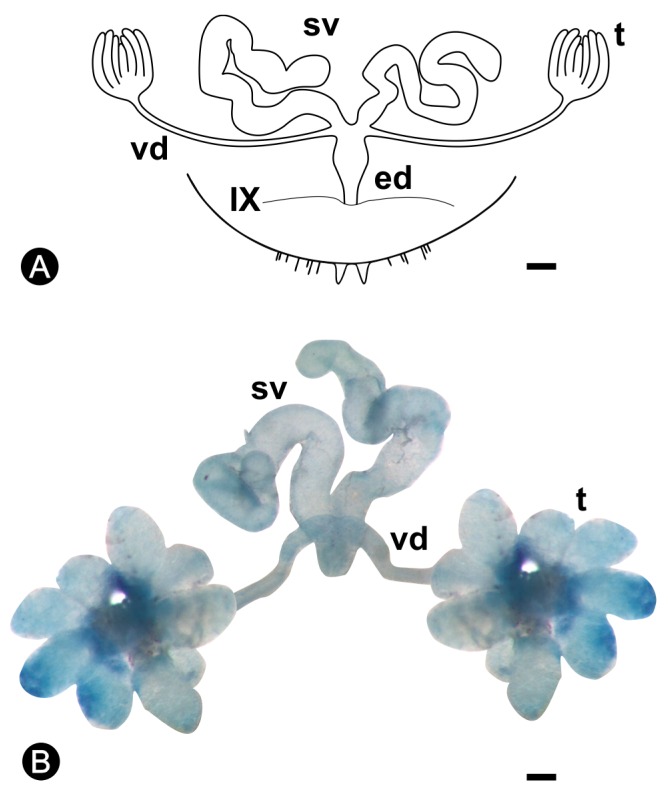
(**A**) Scheme of the male reproductive system of an alate of *Coptotermes gestroi*. (**B**) Total mounting of the male reproductive system of a 6-year king of *C. gestroi*. Staining: methylene blue. ed, ejaculatory duct; sv, seminal vesicle; t, testis; vd, vas deferens; IX, margin of the ninth sternite. Bar = 100 µm.

**Figure 2 insects-10-00428-f002:**
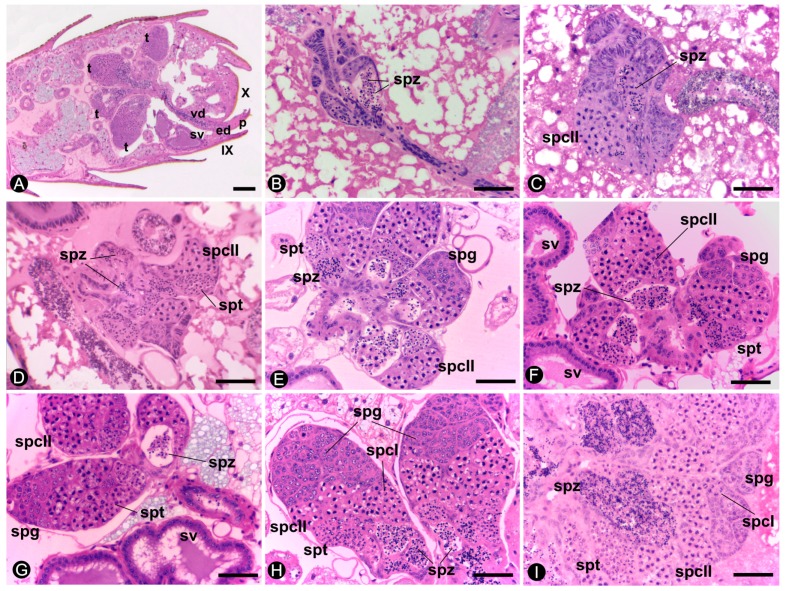
Spermatogenesis and testicular growth in *Coptotermes gestroi*. (**A**) Reproductive system in 4-year king. Note the testicular lobes (t), the vas deferens (vd), the seminal vesicle (sv), and the ejaculatory duct (ed) ending in the penis (p). Testicular lobes of: (**B**) N3. (**C**) N5. (**D**) Alate. (**E**) 6-month king. (**F**) 1-year king. (**G**) 2-year king. (**H**) 4-year king. (**I**). Neotenic. Hematoxylin–eosin staining. Vas deferens, (vd); spermatogonia, (spg); spermatocytes I, (spcI); spermatocytes II, (spcII); spermatids, (spt); spermatozoa, (spz). IX, 9th sternite; X, 10th sternite. Scale bar = 50 µm.

**Figure 3 insects-10-00428-f003:**
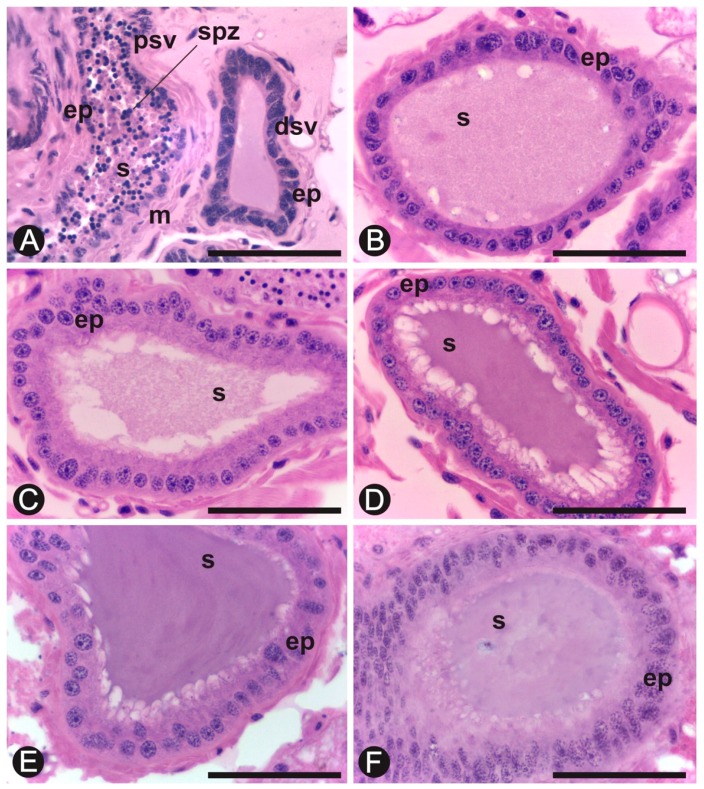
Seminal vesicles of *Coptotermes gestroi*. (**A**) Proximal portion (**psv**) containing spermatozoa (spz) and distal portion (dsv) of the seminal vesicle in alate. (**B**) 6-month king. (**C**) 1-year king. (**D**) 2-year king. (**E**) 4-year king. (**F**) Neotenic. (ep), vesicular epithelium; (m), musculature; s, secretion. Staining: hematoxylin–eosin. Bar = 50 µm.

**Figure 4 insects-10-00428-f004:**
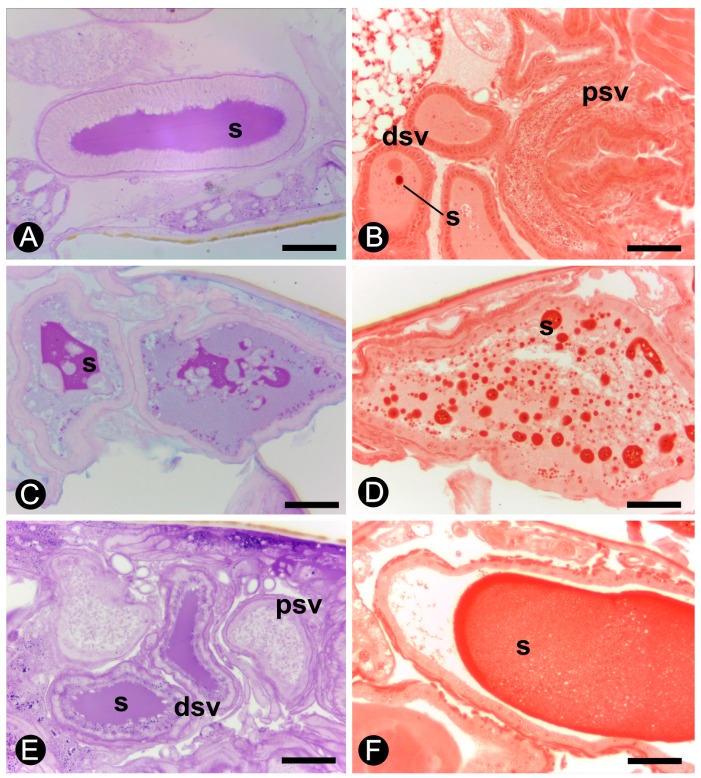
Histochemistry of the seminal vesicles of *Coptotermes gestroi*. (**A**) and (**B**) Alate. (**C**) and **(D**) 6-month king. (**E**). 4-year king. (**F**). 1-year king. Staining: A. and E.: PAS; C.: PAS/Alcian blue; B., D. and F.: xylidine Ponceau. (dsv), distal portion of the seminal vesicle; (psv), proximal portion of the seminal vesicle; **s**, secretion. Scale bar = 50 µm.

**Figure 5 insects-10-00428-f005:**
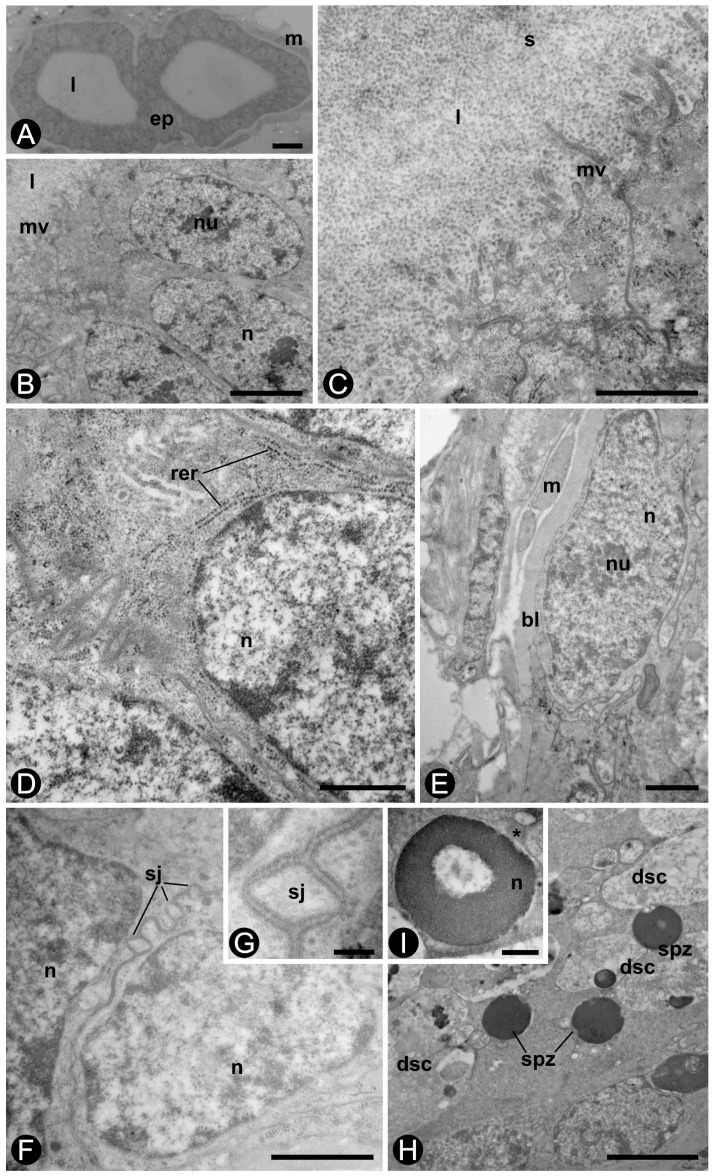
Seminal vesicles of *Coptotermes gestroi* alate (distal portion, **A**–**D**) and king (proximal portion, **E**–**I**). (**A**) Semi-thin section through seminal vesicle. Staining: methylene blue. Scale bar = 10µm. (**B**) Epithelial cells and microvilli (mv). Scale bar = 2 µm. (**C**) Microvilli and flocculated secretion (s) in the lumen (l). Scale bar = 1 µm. (**D**) Profiles of rough endoplasmic reticulum (rer) in the cytoplasm of an epithelial cell. Scale bar = 500 nm. (**E).** Smooth muscle (m) surrounding seminal vesicle. Scale bar = 1 µm. (**F**) Epithelial cells. Note the septate junction (**sj)** adhering adjacent epithelial cells. Bar = 1 µm. **(G).** Septate junction (sj). Scale bar = 1 nm. (**H**) Lumen of seminal vesicle. The spermatozoa (spz) lie among the degenerative sex cells (dsc). Scale bar = 2 µm. (**I**) Spermatozoon with two regions in the nucleus of different electron-densities. Scale bar = 500 nm. (ep), vesicular epithelium; (bl), basal lamina; (n), nucleus, (nu), nucleolus, *****, centriole region.

**Table 1 insects-10-00428-t001:** Mean ± standard error (S.E.) of the testicular area (in μm^2^) in the male reproductives of *Coptotermes gestroi*. Different letters mean statistical differences (Tukey honestly significant difference (HSD) test). Presented means are of the actual data, and analysis was performed using log-transformed data.

Stage	Number of Individuals (n)	Testicular Area (μm^2^)
Nymph 3	3	9618.00 ± 129.61 a
Nymph 4	3	10204.30 ± 217.90 a
Nymph 5	3	20893.45 ± 155.91 b
Alate	6	28478.40 ± 446.12 c
Neotenic	3	34415.66 ± 121.50 d
6-month king	3	38402.90 ± 743.36 e
1-year king	6	40024.95 ± 171.60 e
4-year king	3	76808.26 ± 421.89 f

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
