# Peer review of "A Glycoproteinaceous Secretion in the Seminal Vesicles of the Termite Coptotermes gestroi (Isoptera: Rhinotermitidae)"

_insects, 2019, doi:10.3390/insects10120428_

Round 1
Reviewer 1 Report
Summary (aims and main contributions)
As some of the longest-lived insects, termite kings and queens mate regularly throughout their multi-year (or multi-decade) lives. They differ from most other insects in their lack of accessory glands and sclerotised genitalia, as well as the shape of their sperm. This manuscript uses histological techniques to investigate the reproductive structures of male Coptotermes gestroi at different points in development, with a particular focus on the seminal vesicles. This is an important and interesting contribution because a) little is known about termite reproduction, particularly the male contribution, in spite of their ecological and economic importance and b) systematic study of termite reproductive structures, associated compounds, and mating strategies can further our understanding of (eu)social evolution and male-female conflict and cooperation. The manuscript provides the second characterisation of secretions in the seminal vesicles of termites. The differing developmental trajectories of imago and neotenic lineages is particularly interesting, however functional neotenic kings are unfortunately missing from the study. This manuscript is so similar to a paper on Silvestritermes by the same authors, that it is not clear to me why the two were not published as a single study. The data presented here are interesting and important and should be published, but to be considered novel and significant requires some “added value” beyond simple replication of Laranjo 2018 (see below for suggestions).
Major comments
Introduction: The Introduction jumps from specifics of termite reproductive systems to information about Coptotermes gestroi and its development, then back to specifics of termite reproductive systems, followed by one sentence broadly describing the study. There are hints but no clear thread through the Introduction to show the reader how this study builds on what is already known or how it will fill gaps in our knowledge. Together with some background information currently in the Discussion, the Introduction could be reorganised something like: 1) Termite males differ from other insects in… 2) Comparison of male reproductive development, anatomy and physiology between termite taxa… 3) The rhinotermitid Coptotermes gestroi is known to… 4) We expand on this knowledge base by…
Methods/Discussion: Methods are appropriate to the question. Non-functional neotenics are unfortunately missing from the morphology and TEM aspects and are not mentioned in some of the histochemistry Results although the Methods suggest they were included, eg protein in the sv lines 186-190. It is unfortunate that no functional neotenic kings or N6 were included in the study, but I imagine it is because none were found at the time of collection. It could be a great improvement if it would be possible to include them (perhaps there are now some functional neotenics in the orphaned lab colonies?), but if not, neotenic and primary king development should at least be compared/contrasted in the discussion based on the results from this study and the existing literature.
Results: There is overlap between observations made using techniques described in sections 2.2 – 2.4, such that the same topic is covered in different sections of the Results (everything in lines 125–130; 136–144 is morphology, even though some of the observations are visualised on histological sections) and are occasionally repeated (compare lines 151 and 220). The Results would present a more unified story if data from all techniques were consolidated under thematic headings such as Morphological Overview, Testes and Sperm, Vas Deferentia, and Seminal Vesicles.
I found it
difficult to keep track of comparisons across developmental stages
and which stages were examined for each aspect. This could be
improved by adding a summary table either in the body of the paper or
as supplementary material; while reading, I constructed one with a
row for each developmental stage (N3-6y king, nf neotenic), and
columns for presence of mature sperm in testes and seminal vesicles
(sv), and histochemistry of the sv contents with subcolumns for
neutral polysaccharides, acid polysaccharides, and appearance of
proteins (granular, sparse; granular, many; homogeneous), and would
have also appreciated columns for (relative) size of testes and sv.
The latter should be possible from images of either the morphological
specimens or the sections. Reorganising Figure 4 and labelling Figure
5 (see specific comments below) would also make comparisons easier
for the reader.
Discussion: This study is very similar to Laranjo et al 2018; because two of the journal criteria are novelty and significance, it is important to take this work beyond the previous paper. The first half of the Discussion basically says there is nothing new about the results it highlights. With that in mind, much of the background information could go to the Introduction and the first two paragraphs be summarised in a couple sentences.
I think the best way to increase the novelty and significance of the manuscript would be to include comparisons with other taxa to highlight what may be a “general Isoptera pattern” and what is unique to Rhinotermitidae or Coptotermes (or derived Rhinotermitidae + Termitidae). There are several recent studies exploring reproductive development in male termites; the manuscript cites Laranjo et al 2018, but misses some other relatively recent papers, for example Barbosa and Constantino 2017 (Serritermes), Su et al. 2015 (Reticulitermes labralis), Oguchi et al. 2016 (Hodotermopsis). Taken together with the older literature, all these results suggest there may be something very interesting happening in the reproductive systems of the derived Rhinotermitids + Termitidae. The discussion of the secretion in the seminal vesicles is important but restating the Results could be minimised (most of paragraph 267–275) to allow the manuscript to elaborate on the implications of this finding, and similarities and differences between termites, their closest relatives, and other social insects. The take-home message could be stated more definitely and clearly: the structure and function of the seminal vesicles and their fluid is different in derived Rhinotermitidae + Termitidae than in the basal taxa, which may have something to do with a) evolution of fixed sterile castes (including testes development in the nymphal line vs workers/pseudergates and soldiers), b) if and how replacement reproductives are generated, and c) longer lifespans of (primary) reproductives. There may be parallels to proteins known from accessory glands and sv of other social and non-social insects, but the expectation is that their functions will tend toward protection and support rather than competition and antagonism, due to the well-aligned interests of males and females, with generally low frequency of polyandry, long lifespans of both partners, etc.
Minor comments, by line number
14: “although… occur” sounds a bit odd, as if neotenics never become functional reproductives in this species. Perhaps just delete non-functional from this clause.
21: I believe testes is generally used in insects, not testicles (a term which usually includes associated glands as well as the spermatogenic organs). This should be changed throughout the manuscript.
33: This would be a good place to mention other differences between termites and the general insect pattern, including lack of sclerotised genitalia and accessory glands. This information is vaguely scattered in the abstract and discussion, but is needed here as background. A sentence or two about secretions of the male reproductive system in other insects, particularly social insects, compared to what is known in termites, would round out this paragraph.
41: I was confused by the use of “aerial” to describe C. gestroi nesting habits. I was under the impression that C. gestroi nests below ground or in moist wood and does not build mounds or arboreal carton nests, although it will forage inside living or dead trees and human-built structures. Perhaps there is a more precise word or phase to convey the intended meaning.
68: Please indicate if the alates had already flown or if they were collected as they emerged.
69: The individuals are certainly older than 6 months, 1 year, etc., so it would be more clear to rephrase this something like: Functional primary kings were collected from laboratory colonies 6 months, 1, 2, 4, and 6 years after colony establishment from wild-caught dispersing imagos.
70: Please include information about how the neotenics were recognised as non-functional vs functional (swelling of abdomen or other characteristics? Or were they assumed to be non-functional because the primary reproductives were found?) and indicate from which larval or nymphal stage they were derived.
125–130; 136--144: According to line 75, nymphs were not included in the morphological study. It seems reasonable to include them in this statement, as the structures can be discerned in other aspects of the study, but by that logic, neotenics should also be mentioned. It would be helpful to add quantitative comparative information on the development of testes and sv here; they can be easily estimated from photographs. Lines 141–144 start to go in this direction. This would help answer various questions: Do all testicular lobes in an individual grow at the same rate? Is the growth pattern and eventual size and shape of testes and sv different between the alates and primary kings compared to non-functional and functional neotenics? Are there differences between non-functional neotenics depending on the nymphal stage from which they are derived?
148: In the context of this sentence, I’m not quite sure what is meant by “and according to the developmental stage of the male life”, particularly since the samples represent divergent pathways rather than a linear developmental progression.
166: “pairs and individualized” seems contradictory, so perhaps there is a better word to convey the intended meaning.
166-168: This sounds as if mature sperm are present in the testes but do not descend to the sv of N3–5 (N6 unfortunately not examined). Does this mean that sperm are found in the sv of swarming alates and non-functional neotenics, as well as primary kings? If there are sperm in the sv of nf-neotenics, that raises the question of how non-functional they really are!
174-175: The last sentence here belongs to the next paragraph.
189, 199: This seems to be contradictory (homogeneous in older kings under H-E staining and flocculated in all in TEM).
196–204: In this section it is sometimes difficult to know what applies to only one of the stages examined and what is a general pattern. For example, lines 196–7: only imagos are mentioned, but surely kings have smooth muscle in the walls of the sv, too? Or lines 203-204: does this mean there fewer mitochondria in these cells in kings than in imagoes?
297–298: This statement would be stronger if the discussion reviewed when sperm production begins in other neotenic-producing taxa compared to those that rarely (or “never”) produce neotenics. Also relevant to comment 166–168: when do sperm start moving to the sv?
Figures:
Figure 2: Caption should made more concise, something like: Spermatogenesis and testicular growth in Coptermes gestroi. A. Reproductive system in 4y king. <notations exclusive to this panel> Testicular lobes of: B. N3, C. N5, D. Imago, E. 6mo king, F. 1y king, G. 2y king, H. 4y king, I. Non-functional neotenic. Hematoxylin-eosin staining. <notations common to most or all panels> Scale bar – 50um. It looks like the vd is shown in some slides but not annotated?
Figure 4: Comparisons would be easier if the figure was arranged so that PAS was on the right and X-P on the left. Then the top panel would be both imago pictures, middle 6mo king, bottom 4y and 1y king.
Figure 5: Consider adding false colour to the flocculated secretion highlighted in panel C. The logical flow through the figure would be improved if panel H moved to position E (large to small scale in imago, then large to small scale in king). If the figure remains this size and the caption on the next (rather than opposing) page, consider whether there is a way to visually separate the imago and king sections of the figure (a black line? Or thin black lines between panels from each stage and the white line between the stages?).
This figure would also benefit from a more concise caption. Perhaps: Seminal vesicles of Coptotermes gestroi imago (distal portion, A-D) and king (proximal portion, E-I). A. Section through sv. B. Epithelial cells and microvilli. C. Microvilli and flocculated secretion in the lumen. D. Epithelial cell. E. Smooth muscle surrounding sv. F. Epithelial cells. G. Septate junction. H. Lumen of sv. I Spermatozoa. <abbreviations> Size of scale bar noted on each panel.
Author Response
Reviewer 1
Summary (aims and main contributions)
As some of the longest-lived insects, termite kings and queens mate regularly throughout their multi-year (or multi-decade) lives. They differ from most other insects in their lack of accessory glands and sclerotised genitalia, as well as the shape of their sperm. This manuscript uses histological techniques to investigate the reproductive structures of male Coptotermes gestroi at different points in development, with a particular focus on the seminal vesicles. This is an important and interesting contribution because a) little is known about termite reproduction, particularly the male contribution, in spite of their ecological and economic importance and b) systematic study of termite reproductive structures, associated compounds, and mating strategies can further our understanding of (eu)social evolution and male-female conflict and cooperation. The manuscript provides the second characterisation of secretions in the seminal vesicles of termites. The differing developmental trajectories of imago and neotenic lineages is particularly interesting, however functional neotenic kings are unfortunately missing from the study. This manuscript is so similar to a paper on Silvestritermes by the same authors, that it is not clear to me why the two were not published as a single study. The data presented here are interesting and important and should be published, but to be considered novel and significant requires some “added value” beyond simple replication of Laranjo 2018 (see below for suggestions).
Major comments
Introduction: The Introduction jumps from specifics of termite reproductive systems to information about Coptotermes gestroi and its development, then back to specifics of termite reproductive systems, followed by one sentence broadly describing the study. There are hints but no clear thread through the Introduction to show the reader how this study builds on what is already known or how it will fill gaps in our knowledge. Together with some background information currently in the Discussion, the Introduction could be reorganised something like: 1) Termite males differ from other insects in… 2) Comparison of male reproductive development, anatomy and physiology between termite taxa… 3) The rhinotermitid Coptotermes gestroi is known to… 4) We expand on this knowledge base by…
Reply: We restructured the introduction to make it clearer.
Methods/Discussion: Methods are appropriate to the question. Non-functional neotenics are unfortunately missing from the morphology and TEM aspects and are not mentioned in some of the histochemistry Results although the Methods suggest they were included, eg protein in the sv lines 186-190. It is unfortunate that no functional neotenic kings or N6 were included in the study, but I imagine it is because none were found at the time of collection. It could be a great improvement if it would be possible to include them (perhaps there are now some functional neotenics in the orphaned lab colonies?), but if not, neotenic and primary king development should at least be compared/contrasted in the discussion based on the results from this study and the existing literature.
Reply: We reconsidered the term no-functional neotenics, as suggested by the reviewers, because all the results indicate they might be functional. We agreed that we unfortunately could not analyze the N6 nymphal instar.
Results: There is overlap between observations made using techniques described in sections 2.2 – 2.4, such that the same topic is covered in different sections of the Results (everything in lines 125–130; 136–144 is morphology, even though some of the observations are visualised on histological sections) and are occasionally repeated (compare lines 151 and 220). The Results would present a more unified story if data from all techniques were consolidated under thematic headings such as Morphological Overview, Testes and Sperm, Vas Deferentia, and Seminal Vesicles.
Reply: We preferred to keep this section in the form that was first presented, but we changed the specific points to avoid overlap.
I found it difficult to keep track of comparisons across developmental stages and which stages were examined for each aspect. This could be improved by adding a summary table either in the body of the paper or as supplementary material; while reading, I constructed one with a row for each developmental stage (N3-6y king, nf neotenic), and columns for presence of mature sperm in testes and seminal vesicles (sv), and histochemistry of the sv contents with subcolumns for neutral polysaccharides, acid polysaccharides, and appearance of proteins (granular, sparse; granular, many; homogeneous), and would have also appreciated columns for (relative) size of testes and sv. The latter should be possible from images of either the morphological specimens or the sections. Reorganising Figure 4 and labelling Figure 5 (see specific comments below) would also make comparisons easier for the reader.
Reply: We added a table with the testicular area size to quantitatively differentiate the individuals during the post-embryonic development and after the imaginal molt with different aged kings.
Discussion: This study is very similar to Laranjo et al 2018; because two of the journal criteria are novelty and significance, it is important to take this work beyond the previous paper. The first half of the Discussion basically says there is nothing new about the results it highlights. With that in mind, much of the background information could go to the Introduction and the first two paragraphs be summarised in a couple sentences.
I think the best way to increase the novelty and significance of the manuscript would be to include comparisons with other taxa to highlight what may be a “general Isoptera pattern” and what is unique to Rhinotermitidae or Coptotermes (or derived Rhinotermitidae + Termitidae). There are several recent studies exploring reproductive development in male termites; the manuscript cites Laranjo et al 2018, but misses some other relatively recent papers, for example Barbosa and Constantino 2017 (Serritermes), Su et al. 2015 (Reticulitermes labralis), Oguchi et al. 2016 (Hodotermopsis). Taken together with the older literature, all these results suggest there may be something very interesting happening in the reproductive systems of the derived Rhinotermitids + Termitidae. The discussion of the secretion in the seminal vesicles is important but restating the Results could be minimised (most of paragraph 267–275) to allow the manuscript to elaborate on the implications of this finding, and similarities and differences between termites, their closest relatives, and other social insects. The take-home message could be stated more definitely and clearly: the structure and function of the seminal vesicles and their fluid is different in derived Rhinotermitidae + Termitidae than in the basal taxa, which may have something to do with a) evolution of fixed sterile castes (including testes development in the nymphal line vs workers/pseudergates and soldiers), b) if and how replacement reproductives are generated, and c) longer lifespans of (primary) reproductives. There may be parallels to proteins known from accessory glands and sv of other social and non-social insects, but the expectation is that their functions will tend toward protection and support rather than competition and antagonism, due to the well-aligned interests of males and females, with generally low frequency of polyandry, long lifespans of both partners, etc.
Reply: We changed the discussion properly, adding the suggested references and discussing about neotenic reproductive system development. We agreed with the reviewer that the proteins from the seminal fluid might be associated with protection of sperm. We also added a paper by King et al. (2011) who empirically demonstrated it in Apis mellifera.
Minor comments, by line number
14: “although… occur” sounds a bit odd, as if neotenics never become functional reproductives in this species. Perhaps just delete non-functional from this clause.
Reply: done
21: I believe testes is generally used in insects, not testicles (a term which usually includes associated glands as well as the spermatogenic organs). This should be changed throughout the manuscript.
Reply: done
33: This would be a good place to mention other differences between termites and the general insect pattern, including lack of sclerotised genitalia and accessory glands. This information is vaguely scattered in the abstract and discussion, but is needed here as background. A sentence or two about secretions of the male reproductive system in other insects, particularly social insects, compared to what is known in termites, would round out this paragraph.
Reply: done
41: I was confused by the use of “aerial” to describe C. gestroi nesting habits. I was under the impression that C. gestroi nests below ground or in moist wood and does not build mounds or arboreal carton nests, although it will forage inside living or dead trees and human-built structures. Perhaps there is a more precise word or phase to convey the intended meaning.
Reply: We changed the sentence in order to make it clear. The term “aerial” used by us was to describe that some carton nests were found very high above ground, in empty spaces of skyscrapers on seventeenth floor, for example. Though we changed the sentence as is was causing confusion, and removed this term “aerial”.
68: Please indicate if the alates had already flown or if they were collected as they emerged.
Reply: The alates were collected directly from swarming, so they are flown alates (line 71, M&M)
69: The individuals are certainly older than 6 months, 1 year, etc., so it would be more clear to rephrase this something like: Functional primary kings were collected from laboratory colonies 6 months, 1, 2, 4, and 6 years after colony establishment from wild-caught dispersing imagos.
Reply: done
70: Please include information about how the neotenics were recognised as non-functional vs functional (swelling of abdomen or other characteristics? Or were they assumed to be non-functional because the primary reproductives were found?) and indicate from which larval or nymphal stage they were derived.
Reply: The neotenics used in this study were collected from a nest headed by the primary reproductives. The females were analyzed in the study by Costa-Leonardo et al. (2004) and did not show sperm in the spermathecae or terminal oocytes, so they were considered non-functional neotenics. The males, on the other hand, were analyzed many years latter, in this study, and were characterized at first as non-functional because of the females. However, in this revised version, we performed a morphometric evaluation to determine the increase (and stage) of the testes in the males, and the size of the testes of the neotenics was bigger than in alates. Thus, considering the testis size and the presence of sperm in the seminal vesicles we considered the reinterpret the neotenics as functional individuals. We removed all “non-functional” throughout the text.
125–130; 136--144: According to line 75, nymphs were not included in the morphological study. It seems reasonable to include them in this statement, as the structures can be discerned in other aspects of the study, but by that logic, neotenics should also be mentioned. It would be helpful to add quantitative comparative information on the development of testes and sv here; they can be easily estimated from photographs. Lines 141–144 start to go in this direction. This would help answer various questions: Do all testicular lobes in an individual grow at the same rate? Is the growth pattern and eventual size and shape of testes and sv different between the alates and primary kings compared to non-functional and functional neotenics? Are there differences between non-functional neotenics depending on the nymphal stage from which they are derived?
Reply: We agreed and removed the specifications concerning nymphs and alates, because all the males of C. gestroi, including the neotenics, present the same morphological structures. We added a morphometric analysis of the testes of the reproductives, estimated from pictures of the most sagittal sections of each individual. This was not able to be performed with seminal vesicles, because they are convolute tubes, so the sections were irregular and did not enable comparison across the different individuals.
148: In the context of this sentence, I’m not quite sure what is meant by “and according to the developmental stage of the male life”, particularly since the samples represent divergent pathways rather than a linear developmental progression.
Reply: We agreed and removed this part of the sentence to clarify meaning.
166: “pairs and individualized” seems contradictory, so perhaps there is a better word to convey the intended meaning.
Reply: We removed “individualized” to clarify meaning.
166-168: This sounds as if mature sperm are present in the testes but do not descend to the sv of N3–5 (N6 unfortunately not examined). Does this mean that sperm are found in the sv of swarming alates and non-functional neotenics, as well as primary kings? If there are sperm in the sv of nf-neotenics, that raises the question of how non-functional they really are!
Reply: We changed the sentence to clarify meaning. The sperm is found in the testes of N3 and older individuals, but they are absent in the seminal vesicles of nymphal instars and alates. They appear in the sv in neotenics and kings. We added this information in the results.
174-175: The last sentence here belongs to the next paragraph.
Reply: done
189, 199: This seems to be contradictory (homogeneous in older kings under H-E staining and flocculated in all in TEM).
Reply: We changed the “homogeneous in older kings” of the sentence to “homogeneously distributed in the seminal lumen in older kings” to avoid misunderstanding.
196–204: In this section it is sometimes difficult to know what applies to only one of the stages examined and what is a general pattern. For example, lines 196–7: only imagos are mentioned, but surely kings have smooth muscle in the walls of the sv, too? Or lines 203-204: does this mean there fewer mitochondria in these cells in kings than in imagoes?
Reply: We agreed with the reviewer and removed the “in imagos” from the sentence in lines 196-7. In lines 203-4, the idea was to describe the cellular content, so we removed “in functional kings” because this content is also present in alates.
297–298: This statement would be stronger if the discussion reviewed when sperm production begins in other neotenic-producing taxa compared to those that rarely (or “never”) produce neotenics. Also relevant to comment 166–168: when do sperm start moving to the sv?
Reply: We added a paragraph in the discussion section comparing C. gestroi to a regular neotenic-producing termite species to support the conclusion of lines 297-298. In addition, we also added when the sperm move to the seminal vesicles in C. gestroi.
Figures:
Figure 2: Caption should made more concise, something like: Spermatogenesis and testicular growth in Coptotermes gestroi. A. Reproductive system in 4y king. <notations exclusive to this panel> Testicular lobes of: B. N3, C. N5, D. Imago, E. 6mo king, F. 1y king, G. 2y king, H. 4y king, I. Non-functional neotenic. Hematoxylin-eosin staining. <notations common to most or all panels> Scale bar – 50um. It looks like the vd is shown in some slides but not annotated?
Reply: done
Figure 4: Comparisons would be easier if the figure was arranged so that PAS was on the right and X-P on the left. Then the top panel would be both imago pictures, middle 6mo king, bottom 4y and 1y king.
Reply: done
Figure 5: Consider adding false colour to the flocculated secretion highlighted in panel C. The logical flow through the figure would be improved if panel H moved to position E (large to small scale in imago, then large to small scale in king). If the figure remains this size and the caption on the next (rather than opposing) page, consider whether there is a way to visually separate the imago and king sections of the figure (a black line? Or thin black lines between panels from each stage and the white line between the stages?).
Reply: we kept this figure as in the original first version because we were not able to change it according to this suggestions.
This figure would also benefit from a more concise caption. Perhaps: Seminal vesicles of Coptotermes gestroi imago (distal portion, A-D) and king (proximal portion, E-I). A. Section through sv. B. Epithelial cells and microvilli. C. Microvilli and flocculated secretion in the lumen. D. Epithelial cell. E. Smooth muscle surrounding sv. F. Epithelial cells. G. Septate junction. H. Lumen of sv. I Spermatozoa. <abbreviations> Size of scale bar noted on each panel.
Reply: done
Reviewer 2 Report
Excellent description of male genitalia development in C. gestroi. Images are of very high quality. Do consider defining each developmental stage of the male in precise terms as suggested in the 2nd comment of attached review.

Author Response
Comments were replied in the pdf file.

Reviewer 3 Report
See attached pdf.

Author Response
Report on manuscript insects-513330 entitled “Morphology of the genital apparatus in males of the termite Coptotermes gestroi highlighting the histochemistry and ultrastructure of seminal vesicles”
In this manuscript, authors did a thorough investigation of the male reproductive
apparatus of Coptotermes gestroi. Notably, they document that a glycoproteinaceous
secretion is found in the seminal vesicles, and that the protein part of the secretion rises in older kings. The paper is overall well written, but I have some general and minor comments prior to its publication in Insects.
First, I am quite troubled by the title of the paper as it seems to be double-sided: how does “Morphology…” turns into “highlighting…”? I would suggest changing the title, and maybe mentioning the finding of glycoproteinaceous secretion?
Reply: We changed the title properly, as suggested.
Second, the structure of the introduction is somewhat strange because they first focus on the morphology of the male reproductive apparatus and sperm, then on the model species, and back to the reproductive structure. For me, a more logical structure would be: model species (lines 38-54), then the morphology of male apparatus (lines 32-37, 55-65).
Reply: We reordered the introduction paragraphs as suggested in order to clarify it. This was also a suggestion of the reviewer 2.
Finally, I generally find that the structure of the legends of Figures are somewhat misleading and should be restructured.
Reply: We changed the figure legends properly, as suggested. This was also a suggestion of the reviewer 1.
Minor comments
Introduction
- lines 49-54: Please provide a figure showing the caste differentiation pathways in this species.
Reply: The caste differentiation pathways of Coptotermes gestroi is already published in the paper by Barsotti and Costa-Leonardo (2005), so we do not agree in addition a figure to illustrate that in the introduction of this manuscript, because we did not performed any study on that subjected, only used the information already published in the literature to compare with the individuals we collected in field traps or obtained from lab colonies.
- lines 46-48: The causality is misleading, please rephrase.
Reply: done
- lines 61-65: Sentence is too long, consider rephrasing. The objectives of this study do not appear very clear here, please elaborate in context. Also, give a little more details on the methods used.
Reply: done
Material and Methods
- line 72: Maybe a small table indicating the number of individuals, and their use for histology and/or histochemistry, and TEM, by stage, would be helpful.
Reply: We disagree with the reviewer on adding a table, because it would be most completed by 3s, such as below. So we complemented the sentence adding the exception for alates and 1-year-old kings that the number of individuals were different from 3. We also added the number of individuals used for morphology and ultrastructure.
- lines 96, 102, 106: Please indicate the expected outcome in the case one compound is present.
Reply: Xylol is a clearing agent of the histological routine, so we modified the sentence to clarify it.
Results
- Figure 1A: Please add t. Also indicate in the margin the sternite # where the ed is located.
Reply: done
- Figure 2 (lines 156-158): Please clarify the legend for items. D-G are the same details as for C?
Reply: done
- Figure 3 (lines 177-179): Please clarify the legend for items. B-F only shows the dsv? Therefore, is there not an issue for the scale between A and B-F?
Reply: We modified it properly, following suggestions by reviewer 1
- lines 188 and 193: 1- or 4-year-old king? Please clarify.
Reply: 1-year-old king. We properly corrected it.
- Figure 4 (lines 191-194): Please clarify the legend for items, and association to staining.
Reply: We changed the figure 4 order as suggestion made by reviewer 1, as well as the legend.
- Figure 5H: epz is spz. Please correct.
Reply: done
- line 201: Alate imago is not yet a reproductive sensu stricto. Please be more specific.
Reply: We changed both “reproductives” by both “alates and kings”.
- line 204: What about male alates?
Reply: We replaced the word “imago” by alates or kings to be more specific.
Discussion
- line 272: Compare with the situation in Silvestritermes euamignatus
(https://doi.org/10.1016/j.jcz.2017.11.015) related to reproductive maturation. What
difference in their overall mating biology or frequency of copulation could explain such a difference? Is there any data on the duration between successive copulation between king and queen in C. gestroi? Are there any report of changes in secretions in the seminal vesicles in other insects related to reproductive maturation?
Reply: Unfortunately, there is no record of copulation frequency in neither species nor, to our knowledge, on seminal secretions and reproductive maturation. However, we changed the manuscript properly to improve comprehension.
Round 2
Reviewer 1 Report
The manuscript is much improved! A few minor comments on the new table:
Table 1 would be improved by the inclusion of the number of individuals contributing to each mean, either in an n= column or in the caption; I think it is n=6 for two categories and n=3 for the rest, but it would be nicer for the reader to not have to go back to the methods to check. I would prefer the units of the area of the testis be included in the column header as well as in the caption, and a note in the caption that the means are of the actual data but the analysis was done on transformed data. The column heading "Individual" should be changed to something like "Stage" or "Category".
Author Response
Reply - Reviewer 1
The manuscript is much improved! A few minor comments on the new table:
Table 1 would be improved by the inclusion of the number of individuals contributing to each mean, either in an n= column or in the caption; I think it is n=6 for two categories and n=3 for the rest, but it would be nicer for the reader to not have to go back to the methods to check.
Reply: done. We added an n=column, as suggested.
I would prefer the units of the area of the testis be included in the column header as well as in the caption, and a note in the caption that the means are of the actual data but the analysis was done on transformed data.
Reply: done
The column heading "Individual" should be changed to something like "Stage" or "Category"
Reply: Done, we replaced "individual" by "stage".